# Association of the Inferior Alveolar Nerve Position and Nerve Injury: A Systematic Review and Meta-Analysis

**DOI:** 10.3390/healthcare10091782

**Published:** 2022-09-16

**Authors:** Yangjie Li, Ziji Ling, Hang Zhang, Hanyu Xie, Ping Zhang, Hongbing Jiang, Yu Fu

**Affiliations:** 1Jiangsu Province Key Laboratory of Oral Diseases, Nanjing Medical University, Nanjing 210029, China; 2Jiangsu Province Engineering Research Center of Stomatological Translational Medicine, Nanjing Medical University, Nanjing 210029, China; 3Department of Oral and Maxillofacial Surgery, Affiliated Hospital of Stomatology, Nanjing Medical University, Nanjing 210029, China

**Keywords:** inferior alveolar nerve canal, inferior alveolar nerve injury, third molar, tooth extraction

## Abstract

Background: We aimed to compare the relationship between the buccal and lingual positions of the inferior alveolar nerve canal (IAC) relative to the lower third molar (LM3) and the rate of the inferior alveolar nerve (IAN) injury. Methods: A systematic search was performed in the following databases: PubMed, Cochrane Central Register of Controlled Trials (CENTRAL), Web of Science, and Journals@Ovid. No language or publication status restrictions were set. The publication year was set from 2009 to 2021. The process of meta-analysis was performed by Review Manager software (Cochrane Collaboration). Results: A total of 1063 articles were initially searched and full texts of 53 articles were read, and 11 satisfactory articles were found. There was a statistical difference between the rate of IAN injury and the lingual position and buccal position of the IAC relative to the LM3 roots (OR, 4.96; 95% CI, 2.11 to 11.62; *p* = 0.0002), with high heterogeneity (*p* = 0.001, I^2^ = 65%). Conclusion: A statistical difference was found in the rate of IAN injury between cases where the IAC was positioned buccally and lingually of the LM3 roots. The IAC was at a relatively higher risk of damage in third molar extraction when it was located on the lingual position of the LM3 roots.

## 1. Introduction

Surgical removal of the lower third molar (LM3) is a common oral and maxillofacial surgery [1]. Inferior alveolar nerve (IAN) injury is an uncommon but serious nerve complication that often results in lawsuits from some patients following the extraction of the LM3. IAN injury can result in serious abnormal consequences, such as paresthesia of the lower lip region, numbness, and decreased quality of life [2,3]. The rate of IAN injury after LM3 extraction varies from 0.4% to 8% [1,4,5]. The reported rate of permanent IAN injury is no more than 1% [4,5]. The age of the patient, inexperience of the surgeon, horizontal angulation, deep impaction, and position of the inferior alveolar nerve canal (IAC) to the LM3 roots have been recognized as risk factors for IAN injury [6,7]. 

Surgeons have been devoted to reducing the incidence of IAN injury by identifying risk factors [8]. The anatomical relationships between IAN injury and the status of the LM3 roots are major research directions in clinical studies, which focus on the following: (1) one or both sides of the IAC cortical bone interruption, (2) deviation shift of the IAC, (3) darkening of the roots, (4) sudden fracture of the roots, (5) narrowing of the roots, (6) apical ramification of the roots, (7) narrowing of the IAC, and (8) positions of the IAC to the LM3 roots [9,10]. The position of the LM3 roots with respect to the IAC is a good reference for the prognosis of IAN injury [8]. A study with 537 LM3 extractions found that the lingual position of the inferior alveolar nerve relative to the LM3 increased the rate of IAN injury [11]. Hasegawa and his colleagues also revealed that the rate of IAN injury is higher during LM3 extraction in cases with a lingual position of the IAC because the canal is presumed to be sandwiched between the lingual cortical bone and the LM3 roots [9]. Furthermore, in the process of LM3 extraction, the tooth dislocation cannot be performed along the long axis of the tooth and the root may touch the IAN to cause IAN injury, which is followed by symptoms such as hypesthesia, paresthesia, or dysesthesia [11,12]. Anatomical factors including angulation and the level of impaction have a great influence on postoperative complications after the extraction of the LM3 [13]. A study reported that a difference was found in the anatomical structure of the IAN in patients with and without mandibular asymmetry [14].

Based on the coronal view of cone beam CT (CBCT) images, the IAC position relative to the LM3 roots is commonly classified into the following four types: (1) buccal, (2) lingual, (3) inter-radicular, (4) inferior. According to other studies, buccal and lingual positions of the IAC relative to the LM3 roots occupy an important position among the reported causes of complications [9]. Inter-radicular and inferior positions of the IAC relative to the LM3 roots have aroused classification disputes in some articles [4,15,16]. Some meta-analyses have examined the CBCT versus panoramic radiographs used to reduce the incidence of IAN injury. However, no meta-analysis was found to compare the different positions of the IAC relative to the LM3 roots with IAN injury.

The purpose of this review was to compare the relationship between the buccal and lingual position of the IAC relative to the LM3 roots and the incidence of IAN injury in patients with third molar extraction. This may provide some suggestions for surgeons to be more vigilant in some particular cases. We devised a null hypothesis that there was no statistical difference in the rate of IAN injury between cases where the IAC was positioned buccally and lingually with regard to the LM3 roots.

## 2. Methods

This review was conducted according to the Preferred Reporting Items for Systematic Reviews and Meta-Analyses (PRISMA) guidelines [17]. We registered the protocol on PROSPERO with a registration number: CRD42022356334.

### 2.1. Eligibility Criteria 

We included studies in the present meta-analysis according to the following criteria. (1) Type of participants (P): At least one LM3 is required to be extracted regardless of gender, age, race, social position, or economic income in studies. A CT examination has been performed to evaluate the anatomical relationship in all three dimensions. (2) Type of interventions (I): Studies included at least one side of the IAC located in lingual position of the LM3 roots. (3) Type of comparisons (C): Studies included at least one side of the IAC located in the buccal position of the LM3 roots. (4) Outcome (O): Studies that presented the rate of the IAN injury. (5) Type of studies (S): Both randomized and nonrandomized controlled trials, prospective and retrospective cohort studies, and cross-sectional studies were included. The publication date of articles was set from 1980 to 2022.

Exclusion criteria: Letters, reviews, cell experimental studies, case reports, and animal experimental studies were excluded from the analysis.

### 2.2. Primary Outcomes

The number of IAN injuries in each group. 

### 2.3. Electronic Searches

Two investigators (Li and Ling) searched the following databases: PubMed, Cochrane Central Register of Controlled Trials (CENTRAL), Web of Science, and OVID. No language or publication status restrictions were set. The authors applied Boolean operators to link keywords used for searching. An example of a search conducted in PubMed is shown as follows: (((((((((((((((((((((Mandibular Nerve Injuries[MeSH Terms]) OR Injury, Mandibular Nerve) OR Mandibular Nerve Injury) OR Nerve Injury, Mandibular) OR Inferior Alveolar Nerve Injuries) OR Lateral Pterygoid Nerve Injuries) OR Masseteric Nerve Injuries) OR Injury, Masseteric Nerve) OR Masseteric Nerve Injury) OR Nerve Injury, Masseteric) OR Auriculotemporal Nerve Injuries) OR Auriculotemporal Nerve Injury) OR Injury, Auriculotemporal Nerve) OR Nerve Injuries, Auriculotemporal) OR Nerve In-jury, Auriculotemporal) OR Deep Temporal Nerve Injuries) OR Mental Nerve Injuries) OR Injury, Mental Nerve) OR Mental Nerve Injury) OR Buccal Nerve Injuries) AND ((((((((Molar, Third[MeSH Terms]) OR Molars, Third) OR Third Molar) OR Third Molars) OR Tooth, Wisdom) OR Wisdom Tooth) OR Teeth, Wisdom) OR Wisdom Teeth)) AND ((buccal) OR (lingual)) Filters: from 1980 to 2022. We searched with no restrictions regarding language or journal. The other search strategies are listed in Appendix A.

### 2.4. Searching Other Resources 


We searched references in included articles to serve as a supplement.We contacted authors by e-mail regarding unclear data.


### 2.5. Selection of Studies

Two reviewers (Li and Zhang) reviewed the title and abstract of each article independently to decide whether to proceed a step further. They then read the full selected articles meticulously, as can be seen in Figure 1. A third reviewer (Fu) resolved any disagreements if needed. We measured the reviewers’ agreement by calculating the k statistic. If a group of patients was reported in different articles, only the latest article was included.

### 2.6. Data Extraction and Management

Two reviewers (Li and Zhang) independently used data extraction tables to extract the data. We recorded the following information from each article: authors’ names, year of publication, details of participants, relative countries, sample size of interventions and outcomes, and study design. Risk of bias in included studies was assessed.

The process of meta-analysis was performed by Review Manager software, Version 5.3.3.0 (The Nordic Cochrane Centre, The Cochrane Collaboration, Copenhagen, Denmark). The risk of bias was estimated by obtaining a tool in the Cochrane Handbook for Systematic Reviews of Interventions, Version 6.1 (Higgins JPT, Green S, editors, Chichester, UK: John Wiley & Sons, Ltd.) [18]. The tool has the following seven sections: random sequence generation (selection bias), allocation concealment (selection bias), blinding of participants and personnel (performance bias), blinding of outcome assessment (detection bias), incomplete outcome data (attrition bias), selective reporting (reporting bias), and other bias. The judgment of risk of bias may be low, unclear, and high. The Newcastle–Ottawa Scale [19] was used to assess the quality of retrospective cohort studies and case–control studies. The Newcastle–Ottawa Scale has three different sections: 1, selection; 2, comparability; 3, outcome (retrospective cohort studies), or exposure (case–control studies). Total points of the three sections are 4 × (Selection), 2 × (Comparability), and 3 × (Outcome or exposure).

### 2.7. Measures of Treatment Effect and Heterogeneity

It was desirable to include more than 10 available studies when we conducted the meta-analysis. Dichotomous outcomes were estimated using the odds ratio (OR) and 95% confidence interval (CI). *p* value less than 0.05 was considered statistically significant. I^2^ analysis was used to estimate heterogeneity. In case of low statistical heterogeneity (I^2^ < 50%), a fixed effect model was adopted; in case of high statistical heterogeneity (I^2^ > 50%), the random effect model was adopted. According to the value, we chose a hosted-effects or random-effects model. 

### 2.8. Subgroup and Sensitivity Analysis

We planned to conduct a subgroup analysis regarding different relative continents to judge the risk of heterogeneity. In addition, a sensitivity analysis was also a useful method to investigate heterogeneity. Thus, we conducted a sensitivity analysis to serve as a supplement if possible.

## 3. Results

### 3.1. Study Selection

In total, 1063 articles were initially included, comprising 1023 from electronic searches and 2 from other resources; 625 records were retained after duplicates were removed. We screened the 625 articles and finally found 40 articles [4,5,6,8,9,11,15,16,20,21,22,23,24,25,26,27,28,29,30,31,32,33,34,35,36,37,38,39,40,41,42,43,44,45,46,47,48,49,50,51,52] for further assessment. We assessed 40 articles [4,5,6,8,9,11,15,16,20,21,22,23,24,25,26,27,28,29,30,31,32,33,34,35,36,37,38,39,40,41,42,43,44,45,46,47,48,49,50,51,52] by full-text reading, and finally found 11 suitable articles [4,8,9,11,15,16,40,42,46,47,48]. A flow chart of the selection process is shown in Figure 1. The k statistic index, which was more significant than 0.9, showed good agreement between the two reviewers. We included two randomized controlled trials (RCT) [8,42], one case–control study [40], and eight cohort studies [4,9,11,15,16,46,47,48].

Moreover, 29 articles were excluded from among the 40 articles [4,5,6,8,9,11,15,16,20,21,22,23,24,25,26,27,28,29,30,31,32,33,34,35,36,37,38,39,40,41,42,43,44,45,46,47,48,49,50,51,52] that were read as full texts for the following reasons: 1. The articles did not mention the number of IAN injures in each group, which we required for our analysis; 2. The study design included did not fulfil the eligibility criteria (a review study, a case series study, or a letter to the editor); 3. The investigators did not compare the IAN injury in lingually located IAC of the LM3 roots and in buccally located IAC of the LM3 roots. 

### 3.2. Study Characteristics

Each study’s characteristics are presented in Table 1. The publication time of our included articles [4,8,9,11,15,16,40,42,46,47,48] varied from 2009 to 2020. Five studies [8,11,16,40,42] were published after 2015 (one in 2015, one in 2017, two in 2019, and one in 2020). From the perspective of regional sources, two studies [8,42] came from Europe (one from the Netherlands, and one from Denmark), eight studies [9,11,15,16,40,46,47,48] came from Asia (six from Japan, one from China, and one from South Korea), and one study [4] came from South America (Brazil). Two studies [8,11] did not provide detailed age information about the included patients, and other studies [4,9,15,16,40,42,46,47,48] provided information such as minimum ages, maximum ages, and average ages. Matzen, Ghaeminia, and their colleagues [8,42] made it clear that the number of patients with permanent nerve damage was one and five, respectively. Four studies [8,9,42,48] performed regular follow-up visits for more than six months with patients who underwent LM3 extraction (two for 6 months, one for 18 months, one for more than 12 months), two studies [16,47] did not provide information about the follow-up time, and five studies [4,11,15,40,46] followed up with patients with regular contact for less than six months after the extraction of the LM3 (one more than 5 months, four of 7 days).

### 3.3. Quality Assessment

One study [4] classified the IAC position relative to the LM3 roots as (a) buccal; (b) lingual; (c) central. Two studies [15,16] classified the IAC position relative to the LM3 roots as (a) buccal; (b) lingual; (c) inferior or under. Through reading the whole articles, we recognized that these three studies’ classifications did not influence the final outcomes. We did not consider the two positions because of the different classifications of inferior and inter-radicular positions relative to the LM3 roots. 

### 3.4. Risk of Bias in Included Studies

We classified seven [9,11,15,16,46,47,48] of the eight cohort studies [4,9,11,15,16,46,47,48] as being of high quality, and one [4] of eight cohort studies [4,9,11,15,16,46,47,48] as being of moderate quality. The detailed scores that each study obtained are shown in Appendix A.

The case–control study [40] was regarded as a moderate-quality study, as shown in Appendix A.

Two RCTs [8,42] were classified as being of moderate quality. The results are shown in Figure 2.

### 3.5. Quantitative Synthesis

The rate of IAN injury was 17.66% (86 of 487) in cases where the IAC was positioned lingually with regard to the LM3 roots, and the rate of IAN injury was 3.80% (31 of 816) in cases where the IAC was positioned buccally with regard to the LM3 roots. The rate of IAN injury in cases where the IAC was positioned in the lingual position relative to the LM3 roots was higher.

### 3.6. Data Analysis 

As shown in Figure 3, the overall rate of IAN injury was higher in lingually located IAC of the LM3 roots than that in buccally located IAC of the LM3 roots. There was a statistical difference in the rate of IAN injury between cases where the IAC was positioned buccally and lingually relative to the LM3 roots (OR,4.96; 95% CI, 2.11 to 11.62; *p* = 0.0002), with high heterogeneity (*p* = 0.001, I^2^ = 65%). 

### 3.7. Subgroup Analysis

A subgroup analysis was conducted according to the classification of different relative continents in consideration of the existence of racial differences [53]. One study [46] came from South America, two studies [8,42] came from Europe, and eight studies [4,9,11,15,16,40,47,48] came from Asia. There was a statistical difference in the Asian group (OR, 7.10; 95% CI, 4.23 to 11.92; *p* < 0.00001). No statistical difference was found in the European group (OR, 1.88; 95% CI, 0.03 to 131.77; *p* = 0.77). The heterogeneity of the Asian group was low (*p* = 0.57, I^2^ = 0%), and the heterogeneity of the European group was high (*p* = 0.0006, I^2^ = 91%). These results indicated that studies from the European group might be associated with the high heterogeneity of the whole research. Detailed information is shown in Figure 4.

### 3.8. Sensitivity Analysis

A funnel plot was created to visualize the origin of the high heterogeneity, as shown in Figure 5. The funnel plot showed that an article [42] may be the origin. A sensitivity analysis was carried out to judge whether the article [42] caused heterogeneity. After we removed the article [42] from the sensitivity analysis, we found that the heterogeneity changed largely. A statistical difference was found in the rate of IAN injury between cases where the IAC was positioned buccally and lingually with respect to the LM3 roots (OR,7.38; 95% CI, 4.50 to 12.12; *p* < 0.00001), with low heterogeneity (*p* = 0.70, I^2^ = 0%). Details are shown in Figure 6.

## 4. Discussion

This meta-analysis included 11 different types of studies [4,8,9,11,15,16,40,42,46,47,48] to assess the rate of IAN injury after LM3 extraction according to the IAC position relative to the LM3 roots. Owing to the quality of some studies, which had flaws, these trials provided moderate evidence. In our review, we found some evidence to support the effect of anatomical variation between the IAC and the LM3 roots. The rate of IAN injury in the lingual position of the IAC relative to the LM3 roots (17.66%) was higher than the overall rate (0.4% to 8%) in previous studies [1,4,5]. There was a statistical difference (*p* = 0.0002) in the rate of IAN injury between cases where the IAC was positioned buccally and lingually relative to the LM3 roots. We rejected the null hypothesis so that there was a statistical difference in the rate of IAN injury between cases where the IAC was positioned buccally and lingually relative to the LM3 roots. Many studies showed that the rate of IAN injury was higher in cases in which the IAC was positioned lingually with regard to the LM3 roots than in cases where it was located in the buccal position of the LM3 roots [4,8,9,15,16,40,46]. 

Our study analyzed an anatomical factor affecting the rate of IAN injury from a novel perspective, while previous meta-analyses discussed the rate of IAN injury by comparing panoramic radiographs versus CBCT. Preoperative imaging is commonly recognized as a helpful tool for minimizing the risk of IAN injury [54,55,56,57]. Panoramic film is usually used as an initial examination to evaluate the approximate anatomical relationship of the LM3 roots to the IAC. If the LM3 roots are very close or directly in contact with the IAC, clinical doctors tend to apply a CT scan to obtain detailed information, which panoramic films are difficult to provide [58,59]. Although many studies [60,61,62] hold the opinion that CT, especially CBCT, is significantly superior to panoramic radiographs in reducing the adverse reactions of tooth extraction surgery, some researchers present different ideas. Two meta-analyses [55,57] failed to find a statistically significant difference in the rate of IAN injury after LM3 extraction between patients who had accepted preoperative CBCT versus those who had accepted preoperative panoramic film. Another meta-analysis [54] considered the results of several diagnostic studies to evaluate the diagnostic ability of panoramic radiography to forecast the rate of IAN injury based on the darkening of the root. However, these meta-analyses [54,55,57] did not analyze the effect of anatomical factors on IAN injury. We compared the rate of IAN injury in both the buccal and lingual positions of the IAC relative to the LM3 roots to fill the gap in related aspects. The rate of IAN injury in cases where the IAC was positioned in the lingual position relative to the LM3 roots was higher, which was proven by the results of many previous studies [5,8,9,10,15,16,22,36,37,42,46,50]. Then, we conducted a subgroup analysis according to different continents to determine the cause of the heterogeneity. The 11 studies [4,8,9,11,15,16,40,42,46,47,48] were classified into three different groups (South American group, European group, Asian group) according to the countries in which the studies were performed. The European group showed high heterogeneity, but the heterogeneity of the Asian group was low, so we carried out a sensitivity analysis on studies from the European group. When we removed the Matzen et al. study [42], we found that the heterogeneity changed largely. There was a statistical difference (*p* < 0.00001) in the rate of IAN injury between cases where the IAC was positioned buccally and lingually relative to the LM3 roots in the 10 remaining articles. From the above analyses, we proposed that the heterogeneity might result from the Matzen et al. study [42]. The included patients were referred to the study clinic by the general dentist in the study of Matzen et al. The buccal position of the IAC relative to the LM3 roots tends to be easily found compared with the lingual position of the IAC relative to the LM3 roots in intractable cases. The general dentist may tend to introduce these cases, so that it may increase the rate of IANI between cases where the IAC is positioned buccally invisibly. The above behaviors may cause a certain bias and affect the final results.

Some areas need to be improved by follow-up research. As previously mentioned, different studies adopted different classifications of IAC relative to the LM3 roots, which may cause a disturbance compared to other studies. There is no unified standard for users to apply. A universally acknowledged classification of the relation between the position of the LM3 roots and IAC should be conducted. Relations of the IAC to the LM3 roots were usually classified into “buccal”, “inferior”, “lingual”, and “inter-radicular” sides, based on which rules may be formulated [9,16,46]. A gold standard may contribute to the development of associated research. Indeed, different investigators established their own criteria to classify risks before tooth extraction, which made it difficult for the researchers to conduct a consolidation analysis to obtain the analysis results. In addition, the time limit of permanent IAN injury and temporary IAN injury is difficult to determine because the time needed to consider a permanent IAN injury varies. Many investigators used the term “permanent” to define an IAN injury that had not recovered by the time of the patient’s final follow-up visit; lesions that had not recovered 6 months after surgery were very likely to be permanent. Cheung, Valmaseda-Castellón, and their colleagues [2,63] proposed that an IAN injury lasting more than 6 months after surgery was very likely to be permanent. Two patients still showed loss of light touch sensation and abnormal sensation of the lower lip at follow-up after 1 year in a study conducted by Kaori Shiratori and colleagues [48]. For this reason, the term “persistent” may be more appropriate than the term “permanent”, because the investigators could not treat the evolution of the patient’s injury as permanent after the 6-month follow-up. A meta-analysis [55] conducted by Adria and colleagues suggested that an IAN injury that lasts more than 6 months should be considered persistent.

Some studies stated that the lingual position of the IAC and IAN injury had a significant association in LM3 extraction [5,8,9,16,42,64]. The conclusion that the lingual position was considered to carry a higher risk of IAN injury was similar to the results in our review. Several studies proposed that the surgical approach is closely related to a higher rate of IAN injury in cases where the IAC is positioned in the lingual position of the LM3 roots [11,65]. Most operations were managed through the buccal approach, with an elevator located on the buccal side [65]. During this process, the lingual tissue was compressed inevitably and IAN injury still occurred, even if the IAC was not adjacent to the roots [11]. Hasegawa and his team members [9] revealed that the rate of IAN injury during the extraction of the LM3 is higher in cases with a lingual position of the IAC, because the lingual cortical bone and the LM3 are presumed to clamp the LAC on the extraction of the LM3. The position of the IAC relative to the LM3 roots occupied an important position in predicting the rate of IAN injury [16]. Except for this, a mandibular structure located behind the LM3 tooth, called the retromolar foramen or the retromolar canal, contains important nerve and vascular structures [66]. The neurovascular contents of the foramen or canal become more important in medical and dental practice because these elements are vulnerable to damage during the implantation of osteointegration implants, endodontic treatment, and sagittal osteotomy [67]. It is vital to understand this anatomical variation in understanding failed mandibular surgery and the spread and metastasis of infection.

A method called coronectomy has been suggested to reduce the risk of IAN injury in LM3 extraction. A lower rate of IAN injury was reported with LM3 coronectomies [68,69]. It is worth noting that there was one case of root eruption into the oral cavity in a study [70] carried out by Shingo Goto and his colleagues. They postponed the tooth extraction because the LM3 root was still close to the IAC. Two years after surgery, a preoperative dental CT image found that the root was separated from the IAC, and the residual root could then be extracted in a relatively simple manner. The authors expressed their view that the retained root, extracted safely, was an advantage of coronectomy, because of the complete separation of the root from the IAC. The buccolingual thickness of the cortical bone is thinner in females than in males, which may contribute to the higher rate of IAN injury in female patients [70]. Shingo Goto and other investigators planned to actively apply coronectomy to the clinic in female patients when their imaging data presented symptoms indicating a close relationship between the IAC and the LM3 roots [26]. However, Chinese patients did not commonly accept the method because of the complications after coronectomy. The common complications were root migration from the IAC, followed by delayed healing, dry socket, peri-apical infection, root exposure, and re-operation, required in cases with root exposure and peri-apical infection [11]. Therefore, we suggest that coronectomy might serve as an alternative operation method for teeth when the IAC is positioned in the lingual position. Orthodontic extraction was another method suggested to reduce the risk of IAN injury in the extraction of the LM3. Bonetti [71] reported that his team treated more than 80 patients without complications by means of orthodontic extraction. The method was reported to entail a low degree of postoperative edema and reduce the rate of IAN injury [72]. The shortcoming of the method is that it needs a relatively long treatment time and frequent follow-up before the tooth is prepared to be extracted [71].

We recommend using the above methods flexibly in clinical work, especially under certain circumstances—for instance, if the LAC is located on lingual side of the LM3 roots. More high-quality clinical trials are needed for further assessment of the value of different surgical methods in reducing IAN injury after LM3 extraction.

## 5. Conclusions

A statistical difference (*p* = 0.0002) was found in the rate of IAN injury between cases where the IAC was positioned buccally and lingually relative to the LM3 roots. The IAC was at a relatively higher risk of damage in the third molar extraction when it was located at the lingual position of the LM3 roots. Our findings may prove the role of anatomical factors, especially the position of the IAC with respect to the LM3 roots, in IAN injury. As the numbers of patients and cases with IAN injury in the included studies were limited, more large-scale and standard investigations should be considered in further studies to improve the evidence in support of this conclusion. 

## Figures and Tables

**Figure 1 healthcare-10-01782-f001:**
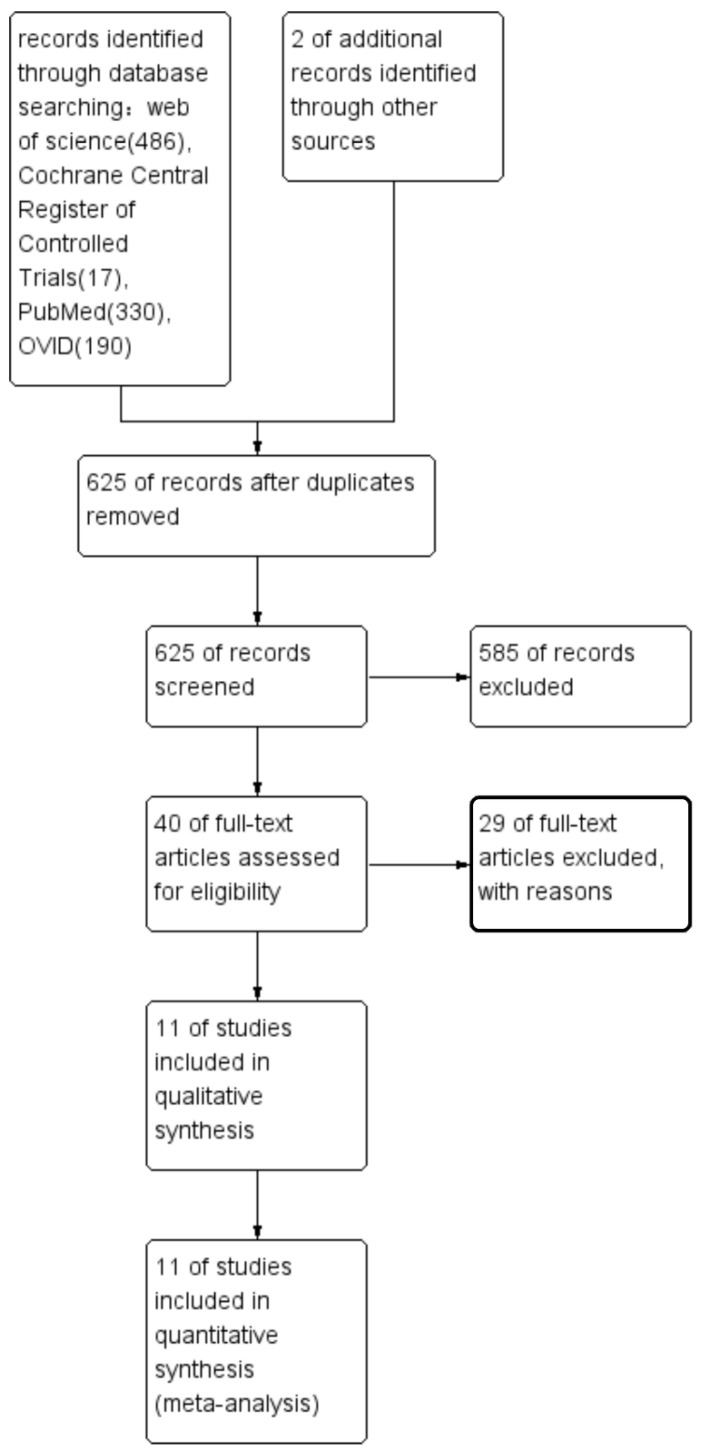
PRISMA flow diagram of study selection process.

**Figure 2 healthcare-10-01782-f002:**
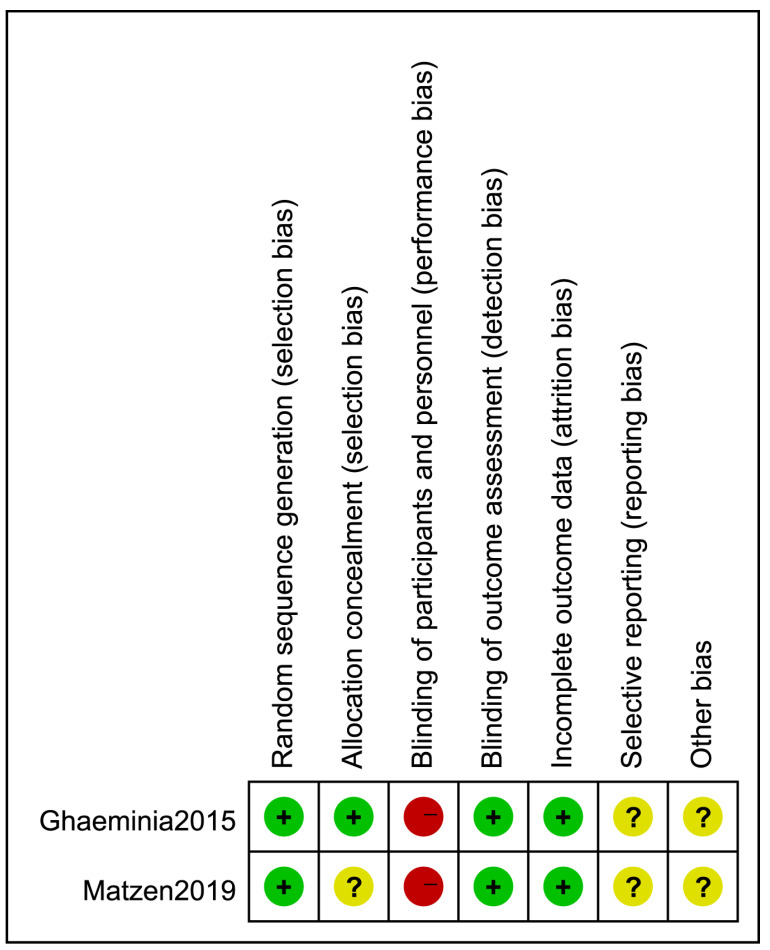
Risk of bias graph: review of authors’ judgments about each risk of bias item for 2 RCT studies. (+) = low risk of bias; (?) = unclear risk of bias; (−) = high risk of bias.

**Figure 3 healthcare-10-01782-f003:**
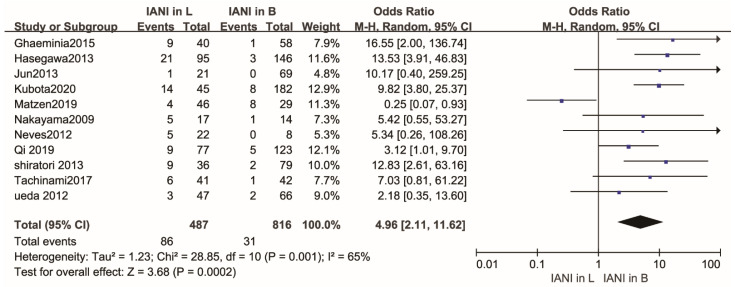
Forest plot of meta-analysis for the rate of inferior alveolar nerve injury in buccal and lingual position of the IAC relative to the LM3 roots after the extraction of the mandibular molars. Abbreviations: IANI, inferior alveolar nerve injury; L, lingual position; B, buccal position; CI, confidence interval; M-H, Mantel–Haenszel.

**Figure 4 healthcare-10-01782-f004:**
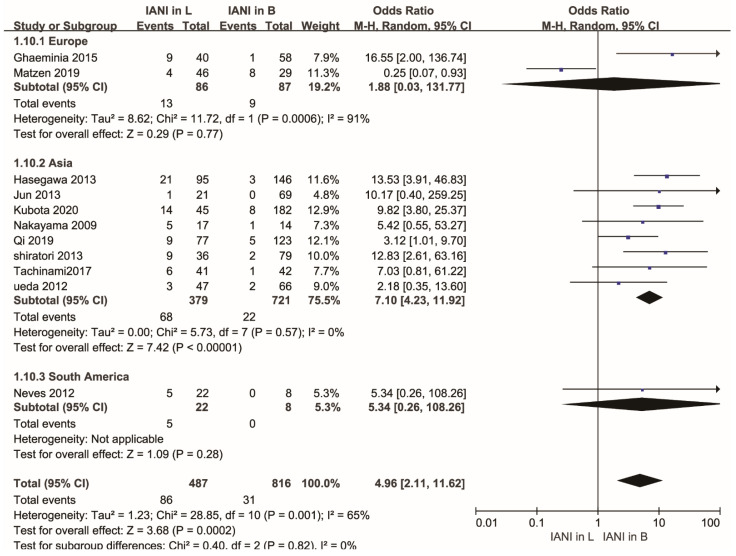
Forest plot of subgroup analysis for the rate of inferior alveolar nerve injury in buccal and lingual position of the IAC relative to the LM3 roots after the extraction of the mandibular molars according to different relative continents. Abbreviations: IANI, inferior alveolar nerve injury; L, lingual position; B, buccal position; CI, confidence interval; M-H, Mantel–Haenszel.

**Figure 5 healthcare-10-01782-f005:**
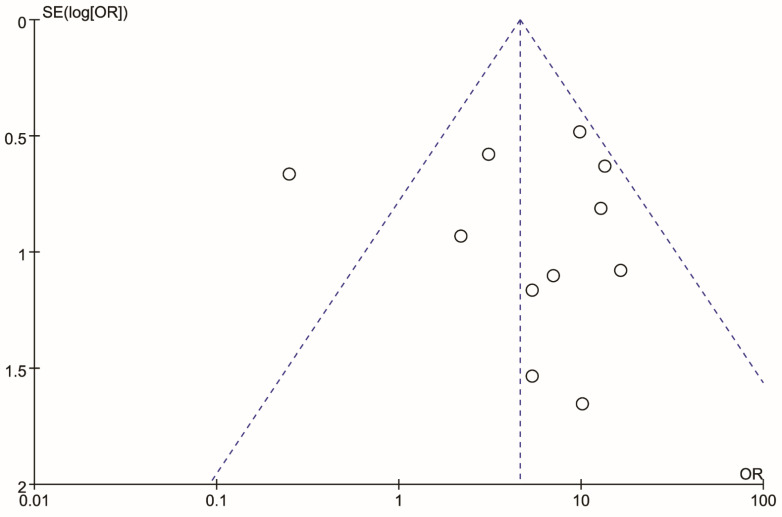
Funnel plot of meta-analysis for the rate of inferior alveolar nerve injury in buccal and lingual position of the IAC relative to the LM3 roots after the extraction of the mandibular molars.

**Figure 6 healthcare-10-01782-f006:**
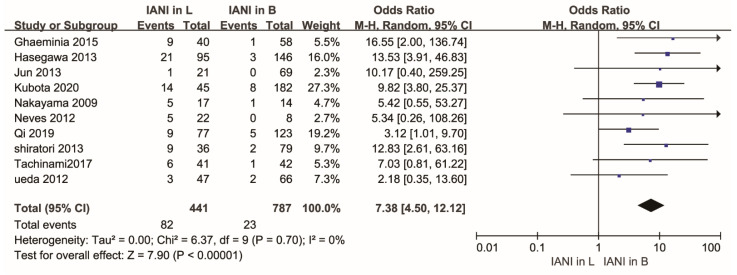
Forest plot of sensitivity analysis for the rate of inferior alveolar nerve injury in buccal and lingual position of the IAC relative to the LM3 roots after the extraction of the mandibular molars. Abbreviations: IANI, inferior alveolar nerve injury; L, lingual position; B, buccal position; CI, confidence interval; M-H, Mantel–Haenszel.

**Table 1 healthcare-10-01782-t001:** Characteristics of the studies included in the meta-analysis.

Authors’ Names	Year	Relative Countries	IANI in L	Sample Size in L	IANI in B	Sample Size in B	Study Design	Age (Years)	Number of P	Follow-Up Time
Ghaeminia	2015	Netherlands	9	40	1	58	an RCT	NM	5	18 months
Hasegawa	2013	Japan	21	95	3	146	a retrospective cohort study	16–71, A36.2	NM	6 months
Jun	2013	South Korea	1	21	0	69	a retrospective cohort study	17–43, A23.9	NM	7 days
Kubota	2020	Japan	14	45	8	182	a retrospective case–control study	23–55, A39	NM	7 days
Nakayama	2009	Japan	5	17	1	14	a retrospective cohort study	18–56, A39.2	NM	7 days
Neves	2012	Brazil	5	22	0	8	a retrospective cohort study	16–55, A26.4	NM	7 days
Tachinami	2017	Japan	6	41	1	42	a retrospective cohort study	17–90, A31.46	NM	NM
Matzen	2019	Denmark	4	46	8	29	an RCT	19–56, A29	1	6 months
Qi	2019	China	9	77	5	123	a retrospective cohort study	NM	NM	>5 months
Ueda	2012	Japan	3	47	2	66	a retrospective cohort study	16–74, A31.8	NM	NM
Shiratori	2013	Japan	9	36	2	79	a prospective cohort study	18–71, 32.5	NM	>12 months

Abbreviations: IANI, inferior alveolar nerve injury; L, lingual position; B, buccal position; A, average age; NM, not mentioned; P, permanent nerve damage.

## Data Availability

Not applicable.

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
