# Peer review of "Association of the Inferior Alveolar Nerve Position and Nerve Injury: A Systematic Review and Meta-Analysis"

_healthcare, 2022, doi:10.3390/healthcare10091782_

Round 1
Reviewer 1 Report
The study is properly designed and technically sound.
The comments I would like to suggest are the following:
- Insert table S1 and table 2 mentioned in the text, but not present.
- The authors may be interested in the information present in these works (DOI: 10.4317/jced.53234; DOI: 10.3390/healthcare9121747;)
- Correct some typos and spelling mistakes.
Best regards
Author Response
Reviewer #1: The study is properly designed and technically sound. The comments I would like to suggest are the following: 1. Insert table S1 and table 2 mentioned in the text, but not present. Response: After examining the reviewer’s comment carefully, maybe our statement was not very clear in the manuscript. Table 2 may not exist in the text. We put the table S1 and table S2 in the supplementary materials with a statement “Supplemental Materials: Additional supporting information may be found online in the Supporting Information section" on lines 384-385. 2. The authors may be interested in the information present in these works (DOI: 10.4317/jced.53234; DOI: 10.3390/healthcare9121747;) Response: We appreciate the reviewer’s constructive comments and thoughtful suggestions. The two studies were helpful to us, so we have added the information in our revised manuscript on lines 262 and 266. 3. Correct some typos and spelling mistakes. Response: We appreciate the reviewer’s constructive comments and we have revised the manuscript and marked the revised portion in red.
Reviewer 2 Report
In my opinion, describing the characteristics of the databases may not be relevant to the manuscript since they are well-known and validated databases. Therefore, it is suggested to delete the corresponding paragraphs or justify why the authors must include definitions.
Figure 2 needs to be improved.
Figure 3 is very complicated to read and interpret. Nevertheless, the data and results are interesting and valuable, but identifying them in the graph is difficult.
The authors make an interesting analysis reporting that the article by Matzen et al. influenced heterogeneity but do not support why this study affects heterogeneity in a deterministic way. Therefore, it is necessary to include a paragraph to this effect in the dissertation.
What was the purpose of dividing the articles by geographic region?
Author Response
Reviewer #2:
- In my opinion, describing the characteristics of the databases may not be relevant to the manuscript since they are well-known and validated databases. Therefore, it is suggested to delete the corresponding paragraphs or justify why the authors must include definitions.
Response: We are sorry for our neglect in this section. We deleted the corresponding paragraphs describing the characteristics of the databases in the revised manuscript.
- Figure 2 needs to be improved.
Response: We appreciate the reviewer’s critical comments and sorry for our neglect in this section. We have improved the Figure 2 in the revised manuscript.
- Figure 3 is very complicated to read and interpret. Nevertheless, the data and results are interesting and valuable, but identifying them in the graph is difficult.
Response: We appreciate the reviewer’s thoughtful suggestion. We divide the four parts of Figure 3. (A-D) into four figures (Figure 3, Figure 4, Figure 5, Figure 6) for a more detailed description in the revised manuscript.
- The authors make an interesting analysis reporting that the article by Matzen et al. influenced heterogeneity but do not support why this study affects heterogeneity in a deterministic way. Therefore, it is necessary to include a paragraph to this effect in the dissertation.
Response: We appreciate the reviewer’s constructive comments and thoughtful suggestions. We included a paragraph to this effect in the dissertation on lines 288-294 in the revised manuscript as follows: The included patients were referred to the study clinic by the general dentist in the study of the Matzen et al. Buccal position of the IAC to the LM3 roots tended to be easy to be found comparing with lingual position of the IAC to the LM3 roots in intractable cases. The general dentist may tend to introduce these cases, so that it may increase the rate of IANI between cases where the IAC was positioned buccally invisibly. The above behaviors may cause a certain bias and affect the final results.
- What was the purpose of dividing the articles by geographic region?
Response: We appreciate the reviewer’s constructive comments and thoughtful suggestions. We stated that a subgroup analysis was conducted according to the classification of different relative continents in consideration of the existence of racial differences [1] on lines 229.

Reviewer 3 Report
The authors could have built a null hypothesis.
The PICO question is missing.
This is a study that should have been registered with prospero - why didn't the authors do it?
What are the exclusion criteria?
The authors should also refer the considered filters.
Why have the authors chose such a short time? “… The publication date of articles was set from 2009 to 2021…”. This can be considered a bias.
99-117 (Materials and Methods) - I see no need to characterize the databases
Pubmed search formule seems incorrect. Only one MeSH term is referred. For instance: what is the purpose of searching “(Buccal Nerve Injury)) OR (Injury, Buccal Nerve)) OR (Nerve Injury, Buccal))”? What is the difference between those 3 segments?
PRISMA 2020 flow diagram should be used (please replace your current flow diagram)
Lines 176-191 – references regarding all the cited articles should be added.
Please add the name of the used risk of bias tools.
“ Our findings may provide new strategies to reduce the risk of the IAN injury.” – this article does not provide any strategies to reduce the risk of and IAN injury. It only provides a study on the prevalence of IAN and on the relationship between buccal and lingual position of IAC relative to the LM3 roots and the incidence of IAN injury. Please rewrite your conclusion according to your initial aim.
Author Response
Reviewer #3:
- The authors could have built a null hypothesis.
Response: We thank the reviewer’s thoughtful suggestions. We thought it is a good idea and we built a null hypothesis in the revised manuscript. We built a null hypothesis that there was no statistical difference in the rate of IAN injury between cases where the IAC was positioned buccally and lingually of the LM3 roots on lines 64-66. One sentence summary of “It rejected the null hypothesis, so that there was statistical difference in the rate of IAN injury between cases where the IAC was positioned buccally and lingually of the LM3 roots.” in the revised manuscript.
- The PICO question is missing.
Response: We appreciate the reviewer’s critical comments. We added the PICOS question on line 72-75 as follows: Participants, interventions, comparators, outcomes, and study design (P.I.C.O.S.) were listed as follows: (1) patients with the LM3 (P); (2) lingual position of IAC to the LM3 roots (I); (3) buccal position of IAC to the LM3 roots (C); (4) IANI (O); (5) clinical studies (S) in the revised manuscript.
- This is a study that should have been registered with prospero - why didn't the authors do it?
Response: We appreciate the reviewer’s constructive comments. We are sorry that we registered with prospero and it has not been approved by the submission date.
- What are the exclusion criteria?
Response: We appreciate the reviewer’s critical comments. We added the exclusion criteria in lines 83-84 in the revised manuscript as follows: Exclusion Criteria: letters, reviews, cell experimental studies, case reports, and animal experimental studies were excluded from the analysis.
- The authors should also refer the considered filters.
Response: We appreciate the reviewer’s constructive comments and thoughtful suggestions. We added the filters in it as follows: Filters: from 1980 – 2022.
- Why have the authors chose such a short time? “… The publication date of articles was set from 2009 to 2021…”. This can be considered a bias.
Response: Response: We appreciate the reviewer’s critical comments. We set the publication date from 1980 to 2022.
7.99-117 (Materials and Methods) - I see no need to characterize the databases
Response: We are sorry for our neglect in this section. We deleted the corresponding paragraphs describing the characteristics of the databases in the revised manuscript.
- Pubmed search formule seems incorrect. Only one MeSH term is preferred. For instance: what is the purpose of searching "(Buccal Nerve Injury)) OR (Injury, Buccal Nerve)) OR (Nerve Injury, Buccal))"? What is the difference between those 3 segments?
Response: We appreciate the reviewer’s constructive comments and thoughtful suggestions. We updated the example of search conducted in PubMed and added the filters as follows: (((((((((((((((((((((Mandibular Nerve Injuries[MeSH Terms]) OR Injury, Mandibular Nerve) OR Mandibular Nerve Injury) OR Nerve Injury, Mandibular) OR Inferior Alveolar Nerve Injuries) OR Lateral Pterygoid Nerve Injuries) OR Masseteric Nerve Injuries) OR Injury, Masseteric Nerve) OR Masseteric Nerve Injury) OR Nerve Injury, Masseteric) OR Auriculotemporal Nerve Injuries) OR Auriculotemporal Nerve Injury) OR Injury, Auriculotemporal Nerve) OR Nerve Injuries, Auriculotemporal) OR Nerve In-jury, Auriculotemporal) OR Deep Temporal Nerve Injuries) OR Mental Nerve Injuries) OR Injury, Mental Nerve) OR Mental Nerve Injury) OR Buccal Nerve Injuries) AND ((((((((Molar, Third[MeSH Terms]) OR Molars, Third) OR Third Molar) OR Third Molars) OR Tooth, Wisdom) OR Wisdom Tooth) OR Teeth, Wisdom) OR Wisdom Teeth)) AND ((buccal) OR (lingual)) Filters: from 1980 – 2022. The charts and a few data changed with the adjustment but it had no impact on the final result and conclusion.
- PRISMA 2020 flow diagram should be used (please replace your current flow diagram)
Response: We appreciate the reviewer’s constructive comments and thoughtful suggestions. We replaced our current flow diagram in the revised manuscript.
- Lines 176-191 – references regarding all the cited articles should be added.
Response: We appreciate the reviewer’s constructive comments and thoughtful suggestions. We added references regarding all the cited articles according to your requirements in the revised manuscript.
- Please add the name of the used risk of bias tools.
Response: We appreciate the reviewer’s constructive comments and thoughtful suggestions. We added the name of the used risk of bias tools (Cochrane Handbook for Systematic Reviews of Interventions, Version 6.1, the Newcastle-Ottawa Scale) in the revised manuscript.
- " Our findings may provide new strategies to reduce the risk of the IAN injury.” – this article does not provide any strategies to reduce the risk of and IAN injury. It only provides a study on the prevalence of IAN and on the relationship between buccal and lingual position of IAC relative to the LM3 roots and the incidence of IAN injury. Please rewrite your conclusion according to your initial aim.
Response: We appreciate the reviewer’s constructive comments and thoughtful suggestions. we rewrote the conclusion because the article did not provide any strategies to reduce the risk of IAN injury on lines 369-370 as follows: Our findings may prove the role of anatomical factors especially the position of IAC to the LM3 roots in the IAN injury.

Reviewer 4 Report
Dear authors!
Thank you for your nice research: as we know problem of IANB injury in some cases is a difficult clinical target to treat.
I have few few questions and hope to get answers.
In the introduction you are speaking about root location but nothing said about type or form of the mandible, its structure and, finally type of the skull. Why didn't you mentioned these anatomical aspects which plays role in complications prognosis.
In Electronic Searches you've added only nerve injury but why didn't you focus your attention at the anatomical variations? Please tell me if you used time period limitation for your search.
Nothing said about retromolar nerve which has very close location to the 3 molar.
In the results data is nicely presented. Were there gender differences in the results? And also does the age of the patient and the condition of the bone tissue matter?
In the conclusion i read "Our findinds may provide new strategies to reduce the 366 risk of the IAN injury" - enlarge this position and add strategies you are speaking about
Author Response
Reviewer #4:
Thank you for your nice research: as we know problem of IANB injury in some cases is a difficult clinical target to treat. I have few questions and hope to get answers.
- In the introductionyou are speaking about root location but nothing said about type or form of the mandible, its structure and, finally type of the skull. Why didn't you mentioned these anatomical aspects which plays role in complications prognosis.
Response: We appreciate the reviewer’s critical comments. We added some contents about the role of mandible and its structure in complications prognosis in line 49-52 as follows: Anatomical factors included angulation and the level of impaction have a great influence on postoperative complications after the extraction of the LM3 [1]. A study had shown that the difference was found in anatomical structure of IAN in patients with and without mandibular asymmetry [2].
- In Electronic Searches you've added only nerve injury but why didn't you focus your attention at the anatomical variations? Please tell me if you used time period limitation for your search.
Response: We appreciate the reviewer’s constructive comments and thoughtful suggestions. We used time period limitations for our search. We updated the example of search conducted in PubMed and added the filters as follows: (((((((((((((((((((((Mandibular Nerve Injuries[MeSH Terms]) OR Injury, Mandibular Nerve) OR Mandibular Nerve Injury) OR Nerve Injury, Mandibular) OR Inferior Alveolar Nerve Injuries) OR Lateral Pterygoid Nerve Injuries) OR Masseteric Nerve Injuries) OR Injury, Masseteric Nerve) OR Masseteric Nerve Injury) OR Nerve Injury, Masseteric) OR Auriculotemporal Nerve Injuries) OR Auriculotemporal Nerve Injury) OR Injury, Auriculotemporal Nerve) OR Nerve Injuries, Auriculotemporal) OR Nerve In-jury, Auriculotemporal) OR Deep Temporal Nerve Injuries) OR Mental Nerve Injuries) OR Injury, Mental Nerve) OR Mental Nerve Injury) OR Buccal Nerve Injuries) AND ((((((((Molar, Third[MeSH Terms]) OR Molars, Third) OR Third Molar) OR Third Molars) OR Tooth, Wisdom) OR Wisdom Tooth) OR Teeth, Wisdom) OR Wisdom Teeth)) AND ((buccal) OR (lingual)) Filters: from 1980 – 2022. The charts and a few data changed with the adjustment but it had no impact on the final result and conclusion.
- Nothing said about retromolar nerve which has very close location to the 3 molar.
Response: We appreciate the reviewer's constructive comments and thoughtful suggestions. We added the relevant contents on lines 329-335 as follows: Except for this, a mandibular structure located behind the LM3 tooth called the retromolar foramen or the retromolar canal contains important nerve and vascular structures[3]. The neurovascular contents of the foramen or canal become more important in medical and dental practice because these elements are vulnerable to damage during the implantation of osteointegration implants, endodontic treatment, and sagittal osteotomy [4]. It is vital to understand this anatomical variation in understanding failed mandibular surgery, spread, and metastasis of infection.
- In the resultsdata is nicely presented. Were there gender differences in the results? And also does the age of the patient and the condition of the bone tissue matter?
Response: We thank the reviewer’s thoughtful suggestions. However, we could not analyse gender differences, the age of the patient and the condition of the bone tissue matter because of lack of original data.
- In the conclusion i read "Our findinds may provide new strategies to reduce the 366 risk of the IAN injury" - enlarge this position and add strategies you are speaking about
Response: We appreciate the reviewer’s constructive comments and thoughtful suggestions. we rewrote the conclusion because the article did not provide any strategies to reduce the risk of IAN injury on lines 369-370 as follows: Our findings may prove the role of anatomical factors especially the position of IAC to the LM3 roots in the IAN injury.
- Baqain Z.H.;Karaky A.A.;Sawair F.;Khraisat A.;Duaibis R., Rajab L.D. Frequency estimates and risk factors for postoperative morbidity after third molar removal: a prospective cohort study. J Oral Maxillofac Surg2008,66, 2276-83.
- Kim J.Y.;Han M.D.;Jeon K.J.;Huh J.K., Park K.H. Three-dimensional assessment of the anterior and inferior loop of the inferior alveolar nerve using computed tomography images in patients with and without mandibular asymmetry. BMC Oral Health2021,21, 71.
- Kumar Potu B.;Jagadeesan S.;Mr Bhat K., Rao Sirasanagandla S. Retromolar foramen and canal: A comprehensive review on its anatomy and clinical applications. Morphologie2013,97, 31-37.
- von Arx T.;Hanni A.;Sendi P.;Buser D., Bornstein M.M. Radiographic study of the mandibular retromolar canal: an anatomic structure with clinical importance. J Endod2011,37, 1630-5.

Round 2
Reviewer 2 Report
In my opinion, the authors addressed all suggestions and comments correctly.
Author Response
In my opinion, the authors addressed all suggestions and comments correctly.
Response: Thanks for the reviewer’s comments and thoughtful suggestions. Your comments have been of great help to us.
Reviewer 3 Report
The authors do not present the PICO question. What the authors wrote does not match. The authors were not able to register the SR in Prosper because there are several flaws or because there is already a record of a SR on the same topic. This registration is essential.
Language is not an inclusion factor, it is a filter.
Author Response
1.The authors do not present the PICO question. What the authors wrote does not match.
Response: We appreciate the constructive comments and thoughtful suggestions. Maybe we didn't present the PICO question clearly, and we presented this question in part of Eligibility Criteria on line 75-83 as follows: (1) Type of participants (P): At least one LM3 is required to be extracted regardless of gender, age, race, social position, or economic income in studies. A CT examination has been performed to evaluate the anatomical relationship in all three dimensions. (2) Type of interventions (I): Studies included at least one side of the IAC located in lingual position of the LM3 roots. (3) Type of comparisons (C): Studies included at least one side of the IAC located in buccal position of the LM3 roots. (4) Outcome (O): studies that presented the rate of the IAN injury. (5) Type of studies (S): Both randomized and nonrandomized controlled trials, prospective and retrospective cohort studies, and cross-sectional studies were included.
2.The authors were not able to register the SR in Prosper because there are several flaws or because there is already a record of a SR on the same topic. This registration is essential.
Response: Our last edition of the registration in Prosper was on August 27, 2022 with a ID of 356334. The website replied that we would register and obtain the registration number in about ten days. The whole process should be completed in the next few days and we hope to give us time to wait for the completion of the registration so as to get the registration number.
3.Language is not an inclusion factor, it is a filter.
Response: We are sorry for our neglect in this section. We removed “(4) Others: No restrictions regarding language or journal were set” on lines 88-89. “We searched with no restrictions regarding language or journal” was added on lines 110-111.
